# The Effects of Tissue Healing Factors in Wound Repair Involving Absorbable Meshes: A Narrative Review

**DOI:** 10.3390/jcm12175683

**Published:** 2023-08-31

**Authors:** Varvara Vasalou, Efstathios Kotidis, Dimitris Tatsis, Kassiani Boulogeorgou, Ioannis Grivas, Georgios Koliakos, Angeliki Cheva, Orestis Ioannidis, Anastasia Tsingotjidou, Stamatis Angelopoulos

**Affiliations:** 1Fourth Surgical Department, School of Medicine, Aristotle University of Thessaloniki, 57010 Thessaloniki, Greece; 2Andreas Syggros Hospital, 11528 Athens, Greece; 3Oral and Maxillofacial Surgery Department, School of Dentistry, Aristotle University of Thessaloniki, 57010 Thessaloniki, Greece; 4Department of Pathology, School of Medicine, Aristotle University of Thessaloniki, 54124 Thessaloniki, Greece; siliaboulog@gmail.com (K.B.);; 5Laboratory of Anatomy, Histology & Embryology, School of Veterinary Medicine, Aristotle University of Thessaloniki, 54124 Thessaloniki, Greece; 6Department of Biochemistry, School of Medicine, Aristotle University of Thessaloniki, 54124 Thessaloniki, Greece

**Keywords:** IL-2, IL-6, CD31, CD34, absorbable meshes, tissue healing, wound repair

## Abstract

Wound healing is a complex and meticulously orchestrated process involving multiple phases and cellular interactions. This narrative review explores the intricate mechanisms behind wound healing, emphasizing the significance of cellular processes and molecular factors. The phases of wound healing are discussed, focusing on the roles of immune cells, growth factors, and extracellular matrix components. Cellular shape alterations driven by cytoskeletal modulation and the influence of the ‘Formin’ protein family are highlighted for their impact on wound healing processes. This review delves into the use of absorbable meshes in wound repair, discussing their categories and applications in different surgical scenarios. Interleukins (IL-2 and IL-6), CD31, CD34, platelet rich plasma (PRP), and adipose tissue-derived mesenchymal stem cells (ADSCs) are discussed in their respective roles in wound healing. The interactions between these factors and their potential synergies with absorbable meshes are explored, shedding light on how these combinations might enhance the healing process. Recent advances and challenges in the field are also presented, including insights into mesh integration, biocompatibility, infection prevention, and postoperative complications. This review underscores the importance of patient-specific factors and surgical techniques in optimizing mesh placement and healing outcomes. As wound healing remains a dynamic field, this narrative review provides a comprehensive overview of the current understanding and potential avenues for future research and clinical applications.

## 1. Introduction

The intricate process of wound healing involves a meticulous sequence of steps encompassing three fundamental phases: inflammation, proliferation, and remodeling [1,2]. During the inflammatory phase, neutrophils, monocytes, and macrophages are activated to eliminate cellular debris and counteract microbial intrusion, thereby averting infections [1,3,4]. The proliferative phase lasts three to 21 days, during which quiescent cells like fibroblasts, keratinocytes, and endothelial cells (ECs), which are involved in the re-epithelization process, proliferate and migrate to the site of injury [4,5]. Growth factors (GFs) like keratinocyte growth factor (KGF), transforming growth factor-β (TGF-β), and vascular endothelial growth factor (VEGF), along with cytokines like interleukin-1 (IL-1) and tumor necrosis factor-α (TNF-α) are also prominently expressed [3,6,7]. VEGF, for instance, instigates angiogenesis, ensuring nutrient supply to emerging tissue and orchestrating the reconstitution of the extracellular matrix (ECM) through synthesizing proteoglycans, collagen III, elastin, and laminin [4,6]. Finally, in the remodeling phase, spanning months to years, ECM undergoes gradual degradation, with type I collagen replacing type III collagen and a reorganization of dermal collagen fibers, processes under the precise regulation of matrix metalloproteinases (MMPs) and their inhibitors (TIMPs) that modulate apoptosis rates and new cell differentiation [3,4,8]. 

Throughout these phases, cellular shape and arrangement alterations are mediated by the modulation of cytoskeletal filaments, encompassing actin networks and microtubules, along with their specific binding proteins [9]. It provides mechanical support essential for determining polarity, inducing proliferation, promoting migration, enhancing differentiation, and maintaining proper functioning of the various types of cells synthesized in the wound bed [10,11,12], potentially orchestrated by the ‘Formin’ family, a protein group that becomes attached to microtubules and actins impacting on their nucleation, polymerization, and stabilization. As a result, different wound healing processes are regulated through these formin groups of proteins influencing cell polarity, chemotaxis, morphogenesis, proliferation, kinesis, proliferation, migration, and phagocytosis [13]. 

Wound healing is also involved in forming and remodeling new tissues through inflammation, which reduces the proliferation and migration of fibroblasts, which is vital for forming new tissues [14,15]. Activation and degranulation of platelets occur instantly after an injury, thus releasing chemokines and GFs (e.g., PDGF), which form a localized fibrin clot and suppress blood loss [3,4]. The spatial and temporal coordination of cytokines with various types of cells is crucial for tissue healing [13]. Wounding promotes changes in the function and dynamics of mitochondria, which modulate the downstream signals that participate in the wound healing process by generating reactive oxygen species (ROS), which affect protein function by regulating gene expression or post-transcriptional modifications [14].

Three categories of meshes are currently available for wound repair: non-absorbable synthetic meshes, absorbable synthetic meshes, and absorbable biological meshes [15,16]. Absorbable meshes serve as scaffolds for regenerative functions such as the deposition of collagen, the promotion of growth of novel tissues leading to neovascularization, and the formation of a new mesothelial layer [16,17]. Absorbable synthetic meshes have advantages like consistent material characteristics, predictable resorption profiles, rapid tissue integration, and the promotion of rapid host bacterial clearance [18]. The classification of bioabsorbable meshes has been divided into three types based on strength loss versus absorption rates. The three categories include long-term meshes having high strength retaining potential but remaining in situ > 18 months; meshes losing strength in a period of 6–8 weeks; medium-term meshes where strengths are maintained for 3–4 months but the absorption rate is high allowing the mesh to be absorbed within a year [19]. For example, the absorbable long-term polylactide mesh (LTS-mesh) demonstrated higher endurance (mechanically stable) and reduced formation of connective tissue than the absorbable short-term polyglactin mesh (PG-mesh) when applied in a standardized rat model of full-thickness abdominal wall defects [20]. Absorbable biological (Strattice, Surgisis, and Tachosil) or absorbable synthetic (Gore^®^ Bio-A^®^ and TIGR^®^) meshes have been employed for the repair of inguinal hernias (IHs) demonstrating lesser chronic pain and promising results in patients with a high risk of infections [16]. TIGR showed enhanced biocompatibility, lesser local tissue effects, better time-dependent mechanical, and enhanced overall performance when compared to a non-absorbable polypropylene mesh on implantation in a sheep model [21] and when used in Lichtenstein repair of lateral inguinal hernias (LIHs), was found to be safe, without recurrent infections, and with lesser pain and discomfort in patients [22]. Gore Bio-A, TIGR Matrix, and Phasix^®^ meshes, when used to prevent or treat small, non-contaminated abdominal wall defects in experimental animals, were observed to be safe with no serious complications [23]. In treating incisional hernias, GORE BIO-A and Phasix meshes were reported to be safe in a surgical environment contaminated with microbes, both postoperatively and over a year later [24,25]. Treatment of paraesophageal hernias using laparoscopic crural reinforcement with synthetic or biological absorbable meshes resulted in an improved safety profile, with a majority of the patients remaining asymptomatic, with a good quality of life, lesser requirement of follow-up surgeries, and diminished long-term recurrence rates [26]. However, it has been noted that the mass of the meshes could substantially impact the occurrence of extended complications during the treatment of inguinal hernias (IHs), as opposed to the classification of the meshes as non-absorbable, partially absorbable, or completely absorbable [27]. 

Furthermore, diabetic individuals often face challenges in wound healing due to compromised blood circulation and reduced immune response. Absorbable meshes, serving as scaffolds, introduce a novel dimension to the healing process [28]. These meshes not only provide mechanical support to the wound site but also act as vehicles for the controlled release of growth factors and cytokines, thereby fostering a conducive environment for tissue regeneration. By influencing aspects like angiogenesis, collagen synthesis, and cell migration, these healing factors within the context of absorbable meshes can significantly expedite wound closure and minimize the risk of infection, offering renewed hope for enhanced recovery in diabetic patients grappling with chronic wounds.

The selection of an appropriate mesh type is guided by patient-specific factors, tissue characteristics, and the nature of the surgery. Optimal mesh integration with surrounding tissues is ensured through precise placement techniques, meticulous fixation, and proper sizing. The critical factors governing mesh biocompatibility and weight are the filament type, tensile strength, and porosity level. Notably, the actual tensile strength employed is lower than assumed. Meshes exhibit enhanced flexibility when lightweight, leading to diminished discomfort. Moreover, greater porosity has proven pivotal in reducing infections and shrinkage. Therefore, the most dependable choice is a lightweight mesh with substantial pores and minimal surface area [29].

Despite the advantages, attention needs to be given to postoperative complications of mesh integration, such as infections and seromas. Infections can manifest around the mesh, giving rise to symptoms such as redness, swelling, pain, and fever. Addressing these infections might necessitate antibiotic treatment and, in severe instances, even mesh removal [30]. Additionally, the formation of seromas can impede wound healing and heighten infection susceptibility [31]. Chronic pain at the mesh site, attributed to nerve irritation or entrapment, is observed in certain patients, thereby highlighting the importance of prudent mesh selection, precise placement, and surgical methodology to mitigate this concern [32]. Despite mesh implementation, the occurrence of wound dehiscence remains possible. However, meticulous tension management, adherence to postoperative directives, and comprehensive patient education regarding activity limitations stand as crucial measures in averting this complication [33].

## 2. Role of Interleukins in Wound Healing 

Interleukins (ILs) are a group of signaling molecules that play a crucial role in the immune response, inflammation, and tissue repair processes, including wound healing. The wound-healing process begins with inflammation, during which immune cells are recruited to the wound site. Interleukins, such as IL-1, IL-6, and IL-8, play a significant role in initiating and regulating the inflammatory response. These cytokines promote the migration of immune cells to the wound site and help in removing debris and pathogens. These interleukins might influence the recruitment of immune cells to the mesh site and contribute to the breakdown of the mesh material. The mesh material itself can trigger an immune response, leading to the secretion of various interleukins and other cytokines. The presence of these interleukins can influence the recruitment of immune cells, the proliferation of fibroblasts, and the remodeling of tissue around the mesh. Proper modulation of interleukin activity is essential to ensure that the healing process occurs without excessive inflammation or fibrosis.

a.IL-2 and tissue healing role.

IL-2 in humans is translated as a 153-amino-acid precursor, then processed to a 15.5 kD, 133-amino-acid long, four-α-helix bundle glycoprotein [34]. IL-2 is a cytokine involved in the signaling pathways associated with the immune system, plays essential roles in several key functions of the immune system, and interacts with multiple cytokines to modulate the generation and activation of immune cells, which may also impact wound healing. IL-2 is majorly produced by CD8+ and CD4+ T-cells in an active state [35]. IL-2 mediates its action by binding to a specific receptor (IL-2R), a heterotrimeric protein expressed on the surface of certain immune cells, such as lymphocytes [34]. The IL-2 signal can be transduced through three signaling pathways: JAK-STAT, PI3K/Akt/mTOR, and MAPK/ERK [36]. It induces activation-induced cell death (AICD) and prevents autoimmune diseases by promoting the differentiation of immature T-cells into regulatory T-cells [30]. It plays a vital role in the differentiation of naive CD8+ T-cells into effector and memory T-cells, thus improving immunity [37], stimulating the differentiation of naïve CD4+ T-cells into T helper cells (Th-cells), and enhancing the cytotoxic activity of both natural killer cells and cytotoxic T-cells [37,38]. IL-2 production by T-cells is promoted by fibroblast growth factors (FGFs)-1 or -2, which are crucial for wound healing and angiogenesis [39]. IL-2 may be a crucial factor in the growth of fibroblasts via autophagy or through wound components derived from damaged wound organelles being digested and reallocated [40]. IL-2 stimulates the maturation of (Interferon-γ) IFN-γ-producing TH1 cells, thus augmenting the release process of IFN-γ, which leads to the production of IL-1 and helps in the wound-healing process [41].

IL-2 levels correlate positively with the percentage of burn wounds [42], act locally and not systemically, and reduced IL-2 levels are preferable at certain stages of the burn healing process [43]. In accordance with these, lower levels of IL-2 were observed at the sites of bone fractures [44], along with changes in the phosphorylation of certain proteins of the downstream IL-2 signaling pathways [45]. Wounds receiving IL-2 treatment demonstrated increased hydroxyproline, indicating increased collagen fiber cross-linking and enhanced ECM deposition [46]. On the contrary, low IL-2 levels and slow action of IL-2-regulated collagen fibers cross-linking in cell proliferation may enhance the wound closure quality [47,48]. Research has reported IL-2 signaling alterations to impact disease pathology that implicates tissue and skin damage, like systemic lupus erythematosus, diabetes mellitus, sarcoidosis, and myocardial infarction [49].

b.Role of IL-6 in tissue healing.

IL-6 acts as a pro-inflammatory, pleiomorphic cytokine, and anti-inflammatory myokine [50]. It is encoded by the IL6 gene located on chromosome 7 in humans [51]. Though IL-6 usually occurs as a 212-amino acid-long [52], 26-kD glycoprotein, its isoforms purified from various tissues have different molecular weights ranging from 19- to 70-kD due to alternate splicing of the IL-6gene and tissue-specific, post-translational modifications like phosphorylation and glycosylation [53,54,55]. IL-6 is produced by almost every cell and tissue type in humans in response to a wide variety of stimulating factors; however, several compounds also inhibit its expression [55]. IL-6 expresses its pleiotropic effects by binding to a specific receptor complex (IL-6R) located either on the membrane of the target cell (mIL-6R) through the “classical pathway” or to the soluble IL-6R (sIL-6R) through the“trans-signalling” pathway; this receptor complex consists of two transmembrane domains: CD126, a ligand binding α-chain, and CD130 or gp130, a signal-transducing β-chain [53,54,55]. The binding of IL-6 to the receptor complex results in either (i) the activation of gp130, which in turn leads to activation of the JAK/STAT/SOCS pathway, or (ii) the activation of the tyrosine phosphatase, SHP2, which in turn activates the RAS/RAF/MEK/MAPK/SOCS pathway [55,56,57]. IL-6 plays a major role in various cell functions: plasma cell development, B-cell differentiation, T-cell proliferation, maturation of cytotoxic T-cells, Ig class switching, hepatic acute phase response, inducing the synthesis of serum amyloid A and C-reactive protein, thrombopoiesis, antimicrobial activity of monocytes and neutrophils, maintenance of bodyweight, and protection against mortality in endotoxin-mediated shock and toxic shock syndrome [56]. 

IL-6 levels are elevated in the inflammatory and proliferative stages of wound healing, during which it involves multiple functions but returns to normal levels in the remodeling stage [58]. IL-6 plays a pivotal role in inducing acute inflammation and is, therefore, necessary for the timely activation of the process [59,60]. IL-6, on being released very early in response to injury, induces the chemotaxis of leukocytes into the wound and the differentiation of macrophages, B-cells, and T-cells, stimulates the growth of keratinocytes, the release of other pro-inflammatory cytokines from macrophages, keratinocytes, endothelial cells, and stromal cells residing in the wounded tissue, inhibition of proliferation of fibroblasts, induction of acute-phase protein synthesis, simulation of hematopoiesis and angiogenesis, and the release of adrenocorticotropic hormone [61,62,63,64].

IL-6 plays an intrinsic role in the acute-phase wound response, may modulate local and systemic post-injury events by being the most persistent cytokine to mediate post-injury complications; and is robustly correlated with adverse clinical events and outcomes after mechanical trauma, burn injury, and elective surgery [55]. On wounding, elevated levels of IL-6 were detected within 24 h of bacterial infection during the initial inflammatory phase, which gradually diminished by the eighth day [65]. IL-6 was the only pro-inflammatory cytokine, the levels of which were persistently elevated post-burn injury; IL-6 levels in the serum peaked during the first few hours after injury and correlated directly with the area of the burn injury, the magnitude of the trauma, the duration of surgery, and the risk of postoperative complications [66]. IL-6 activity in human wound fluids was observed to peak within eight hours of surgery and return to baseline by the third day, a temporal pattern that may suppress the proliferation of fibroblasts in later stages [67]. IL-6 levels in the serum enhanced significantly after either laparoscopic or open IH surgery, more prominently in the latter [53], were highest on day one and fell sharply after the third day [68]. The equilibrium between pro-inflammatory cytokines like IL-6 may induce the transition from the inflammation to the proliferation phase, thus improving the healing process in skin wounds [69]. For instance, in liver transplants for treating chronic liver diseases, IL-6 engages in early graft regeneration and enhances the growth of hepatic tissue by inducing the hepatocyte stem cells to regenerate the liver parenchyma [70,71].

Higher IL-6 levels were observed more in non-healing wounds than in healing wounds [8], during the inflammatory phase in skin wounds in humans [72], and in both the inflammatory and granulating phases in venous leg ulcers [73]. Elevated levels of IL-6 observed in diabetic patients with foot ulcers than those without them indicate that it may play a role in their pathogenesis and development [73]. IL-6 promotes fibrosis due to an improper tissue healing process [58], stimulates the chemo-attraction of neutrophils and mitogenic activity of keratinocytes, which is linked to scar formation [8], and synergistically with hyaluronic acid affects the migration of keratinocytes through the activation of the ERK and NF-kB signalling pathway [74]. Reduced IL-6 synthesis can provide an environment conducive to scarless wound healing, as seen in the lack of inflammation observed in the fetal stages [75].

## 3. Role of CD31

Cluster of Differentiation 31 (CD31), also known as Platelet Endothelial Cell Adhesion Molecule-1 (PECAM-1), is a protein that is encoded by the PECAM1 gene located on chromosome 17 in humans [76]. CD31 is a highly glycosylated, 130 kD protein [77], consisting of six extracellular immunoglobulin-like domains, a 574 amino acid long N-terminal domain, a 19 amino acid long transmembrane domain, and a 118 amino acid long C-terminal cytoplasmic domain [76]. CD31 engages in cell–cell adhesion by interacting with other CD31 proteins present in other cells via homophilic and heterophilic interactions with CD-31 proteins [78,79]. CD31 is located on the surface of several cell types like platelets, monocytes, neutrophils, leucocytes, and certain T-cells and is constitutively expressed on the vascular endothelium [80]. Cell–cell signaling mediated through CD31 activates neutrophils, leukocytes, and monocytes [81]. CD31 also facilitates the migration of monocytes and neutrophils [82], natural killer cells [82], and T lymphocytes [83] through homophilic interactions mediated through endothelial cells. CD31 impacts cellular adhesion in the endothelium, cell transmigration, and diapedesis, resulting in angiogenesis and maintenance of vascular stability in the early stages [84,85,86]. Circulating CD31+ endothelial cells participate in blood vessel formation during wound healing, mediating through inflammation [87]. 

## 4. Role of CD34

CD34 is a transmembrane phospho-glycoprotein encoded by the CD34 gene located on chromosome 1 in humans [88]. CD34 is a member of the single-pass, transmembrane, sialomucin family of proteins; it was first identified in hematopoietic stem cells (HSCs) and is involved in cell–cell adhesion by participating in the attachment of HSCs to bone marrow ECM or directly to stromal cells; expression of CD34 is commonly associated with early hematopoietic and late-hematopoietic stem cells, and other non-hematopoietic, tissue-specific stem cells like muscle satellite cells, epidermal precursors, vascular-associated progenitor cells, endothelial progenitor cells, endothelial cells of blood vessels, masT-cells, dendritic cells, corneal keratocytes, and adipose cells which can be used to identify both newly formed and pre-existing blood vessels; CD34 facilitates the migration of various cell types, especially the chemokine-dependent migration of eosinophils and dendritic cell precursors [89,90,91,92,93,94,95,96]. Human aging can negatively impact adipose tissue-derived CD45−/CD34+/CD133+ progenitor cells availability as their number reduces with age and significantly reduces their angiogenic functional capacity [97]. 

CD34+ mesenchymal cells in the intestinal epithelium are genetically programmed to maintain an inducive environment for intestinal epithelial stem cells (IESCs) at homeostasis and facilitate repair post-injury and inflammation in intestines [98]. CD34+ structures with a vessel-like appearance in the mucosal epithelia and striated muscles were identified in human fetuses [99]. Fibrocytes are innovative blood-endured cells that demonstrate a unique cell surface phenotype (collagen+/CD13+/CD34+/CD45+), differentiate, and rapidly accelerate wound repair and scar formation through rapid migration [100,101]. Identifying CD34+ oral mucosa stem or progenitor cells suggests increased angiogenesis after corrective surgery for cleft lip and palate [96]. CD34+ EPCs were initially observed in human skin wounds after two days. Their numbers enhanced in lesions with increasing wound age, and more than 20 EPCs can indicate a wound age of 7–12 days [102].

## 5. Role of Platelet Rich Plasma (PRP) 

Platelets or thrombocytes, originating from the bone marrow, contain several secretory granules, GFs, and cytokines that, in addition to their main function of homeostasis, can affect inflammation, angiogenesis, migration of stem cells, and cell proliferation [103]. Platelet-rich plasma (PRP) (also known as platelet-rich fibrin matrix, platelet-rich growth factors, platelet concentrate, and autologous conditioned plasma) is the supernatant obtained after centrifugation of whole blood samples to remove red blood cells and consists of a PRP protein concentrate [103]. Activation of the platelets in PRP by thrombin or calcium causes the platelet granules to degranulate and release GFs and cytokines, which influence the microenvironment [104]. Some of the most important GFs released by platelets in PRP include VEGF, epidermal GF (EGF), hepatocyte GF (HGF), fibroblast GF (FGF) -a and b, platelet-derived GF (PGDF) -a and b, transforming GF (TGF)-α and β, insulin-like GF (IGF)-1 and 2, MMP-2 and 9, SDF-1α/CXCL12, andIL-8 [103,104].

PRP, a rich source of signaling molecules like GFs, cytokines, chemokines, and other plasma proteins, demonstrates significant mitogenic, angiogenic, and chemotactic properties that can stimulate the healing of wounds in both soft tissues and joints [104]. These GFs play a crucial role in all three phases, thus ensuring complete wound healing [105]. PRP was initially used to treat thrombocytopenia, then in sports injuries, and nowadays is also employed in cardiac, pediatric, plastic surgery, gynecology, urology, ophthalmology, and dermatology [106,107,108]. Since various protocols are available for preparing PRPs, they could result in PRPs with different levels of bioactive compounds, which may modulate the final extent of wound healing [109,110]. The residual plasma and platelet-bound fibronectin may act as bioactive proteins, which may directly influence the remodeling of the ECM, thus exerting a synergistic effect on the repair of chronic wounds [111]. Hence, all components of blood, like the plasma, platelets, RBCs, and WBCs, have important individual roles in tissue repair, and PRP cannot function alone [112]. Further research on various techniques of PRP preparation, the exact mechanisms of action of GFs, their application in combination therapy, and related clinical trials is required [113]. Since PRP variations regarding the platelet content and the donor are observed, and due to the lack of standardized preparation methods, PRP use has been specific [114].

Autologous PRP demonstrates a great similarity to the natural healing process, is safe, and can be produced as and when required from the patient’s blood [115]. Platelet-derived preparations such as PRP or platelet lysate (PL) may help stimulate regeneration in engineered tissue constructs, and activated PRP has been reported as a potential autologous cell carrier [115]. Due to its potential to stimulate and accelerate the process, PRP is gaining interest in skin wound regenerative therapy [116]. In a clinical setting, PRP and platelet-rich fibrin accelerate healing, thus not only reducing the discomfort of patients but also the probability of adverse outcomes such as infections, poor wound closure, and delay in the formation of sufficiently strong bone for subsequent procedures (such as implants); they may improve long-term outcomes in patients with impaired healing due to diseases (e.g., diabetes, osteoporosis, and atherosclerosis), medications (e.g., steroids), lifestyle choices (e.g., smoking), and aging by supplementing the wound environment to restore proper healing [117]. A systematic review of in vitro, in vivo, and clinical studies highlighted the additive effects of PRF on soft tissue regeneration, augmentation, and wound healing for regenerative therapy in medicine and dentistry [118].

PRP-derived molecules and activated PRP can release various antimicrobial proteins for resolving necrotic tissues and promoting wound healing [119], but in vivo and in vitro studies are required to provide sufficient data for the accurate designing and conducting of RCTs in humans regarding specific pathogens and wound types [120]. For instance, in AIDS patients, PRP may be used to sanitize wounds to induce neovascularization and re-epithelialization and prepare the base and edges of unhealed ulcers for consequent skin grafting procedures and tissue expansion [121]. Activated PRP leads to the production of extracellular micro vesicles, which can fully replicate the pro-healing effects of PRP, suggesting their applicability as an alternative to PRP [122]. The risk factors and contraindications associated with the use of PRP have also been reviewed [123].

Co-culture of human skin fibroblasts with PRP in vivo enhanced the accumulation of type I collagen, MMPs-1, and -2. The G1 cell-cycle regulators-cyclin E and CDK4 may improve wound healing in vitro [124]. Since wounds have a pro-inflammatory environment characterized by high protease activities, which decrease GF levels, PRP, a good source of GFs, is a promising alternative for treating recalcitrant wounds [122], especially in patients with Necrobiosis lipoidica diabeticorum [123]. For treating recalcitrant diabetic foot ulcers and venous foot ulcers, the use of PRP was successful [124], was safe as it does not significantly alter the blood hematology or blood chemistry [125], injections of autologous PRP along with the topical application of PRP gel enhanced wound healing and a reduction in wound size [126] PRP with vacuum-assisted closure dressings were more efficient than conventional dressings [127], homologous platelet-gel (PG) enhanced vascularization and re-epithelialization [128], and resulted in better healing outcomes and lower amputation rates [129]. In the healing of chronic ulcers, the local application of PRP improved the quality of life in patients through effective pain relief [130], combined treatment with enhanced stromal vascular fraction, PRP, and fat grafting demonstrated an enhancement in re-epithelization after regenerative surgery [131], and a novel autologous PRF matrix membrane showed significant potential for applicability [132].

Chronic wounds are unresponsive to conventional treatment methods, are quite common, and pose a challenge to clinicians. A systematic review and meta-analysis of PRP-therapy-based in vitro and in vivo studies reported improved healing of partial or complete, chronic, recalcitrant wounds [133,134], a combination of PRP injections and platelet-derived patches improved healing in patients, especially those with diabetes [135], the topical application of autologous-PRP owing to its antimicrobial properties and tissue-regenerative potential is recommended [136], especially in cases where conventional therapy is not sufficient, or surgery is not possible, local immunity is activated; while the pain and risk of infections are reduced [137,138], it significantly enhanced the re-epithelization process [139], demonstrated a considerable enhancement in the formation of healthy granulation tissue and healing edge; lesser pain, slough, bleeding on touch, discharge, and no superficial or deep infections [140,141,142]. PRP alone or combined with a powdered bioengineered skin substitute was used to synthesize a platelet-rich tissue graft, reducing the wound size and depth [143]. Autologous PRP, PPR-gel, and platelet-gel (PG) are safe with a wide range of applicability as tissue regenerative agents in a variety of postoperative procedures, especially in diabetic patients or those prone to surgical complications and as a replacement for connective tissues, activated PRP and fat tissues serves as a tissue matrix for enabling the cell migration, proliferation, differentiation, and granulation [144]. The short-term use of autologous PRP gel also enhanced the healing process consistently over time [145]. Homologous PRP gel reversed non-healing trends [146], induced granulation tissue formation, and reduced the size of cutaneous wounds [147]. Topical therapy with PG may be considered an adjuvant treatment to enhance the healing of cutaneous ulcers due to the formation of the granulation tissue in the initial stages, followed by complete re-epithelization [148]. As reported in a comprehensive review, PRP enhanced the healing of refractory pressure injuries and reduced the length of treatment and pain without any complications, all of which improved the quality of life in patients [149]. In a unique analysis, applying an algorithm developed by the study group before using PRP enhanced the number of successfully healed wounds, ensured that a higher proportion of acute skin wounds did not turn problematic, and allowed more predictable skin healing patterns [150].

Since PRP’s application method has not been standardized, and the identification of the optimum conditions is complete, more controlled clinical studies are required, as only a few reports suggesting a positive role of PRP in the healing of burns are available [151,152]. The healing rate was prominently enhanced, and the healing time was markedly reduced in PRP-treated burn wounds [153]. PRP improved the healing of tendons, ligaments, muscles, and bones, and hence has been applied in treating sports-related injuries [154,155]. In systematic reviews, the efficacy of PRP in various musculoskeletal pathologies like tendinopathies, early osteoarthritis, and acute muscle injuries [156] and when applied to the bone-tendon interface during arthroscopic rotator cuff repair and wound healing has been addressed [157].

## 6. Role of Adipose Tissue-Derived Mesenchymal Stem Cells (ADSCs)

Stem cells like mesenchymal stem cells (MSCs) improve wound healing by expediting angiogenesis and re-epithelialization, leading to granulation tissue development. MSCs have a substantial role in the process mediated through paracrine interactions, reducing wound inflammation and thus augmenting wound closure. As a result, ECM remodeling occurs, facilitating normal skin development and indicating a good therapeutic target [158,159]. MSCs isolated from the umbilical cord (fetal) and bone marrow (adult) tissues have been employed for the treatment of acute and chronic skin wounds [160]. Human adipose-derived stem cells (hADSCs) play crucial roles in the healing of cutaneous wounds by promoting cell proliferation, migration, differentiation, angiogenesis, matrix reconstruction, and regulation of the inflammatory response and collagen remodeling [161].

Stem cell classification can be based on their origin, such as (a) embryonic, (b) fetal, (c) adult, and (d) induced pluripotent that can be designated as embryonic and adult mesenchymal stem cells (MSCs). Human embryonic stem cells (hESCs) and fetal mesenchymal stem cells (hfMSCs) are difficult to culture due to lack of supply and ethical concerns. The major limitation of employing induced pluripotent stem cells (iPSCs) is the laboratory procedure used to induce their differentiation into specific cell types requires certain disease action [162]. Due to these reasons, adult human stem cells have huge potential in clinical practice and basic research. From a clinical perspective, obtaining stem cells is both cost and time-intensive and involves the risk of contamination and loss. The source of stem cells should be easily accessible, which should cause minimal discomfort and provide enough cells without those time and cost-intensive processes. The sources of adult stem cells include the muscle, the bone marrow, the blood, the epidermis, the brain, the liver, and most recently, the adipose tissue [163,164,165,166].

In vitro and in vivo studies suggest that ADSCs are classified as mesenchymal cells having the capacity of self-renewal and differentiation into tri-germline ages (endoderm, mesoderm, and ectoderm) such as adipocytes, chondrocytes, myocytes, cardiomyocytes, hepatocytes, neurocytes, osteoblasts, vascular endothelial cells, and pancreatic cells [167,168] using specific triggers available in the laboratory [169,170,171,172,173]. The prominent benefits of ADSCs in comparison with other MSCs are easy availability and abundant sources for isolation, convenient tissue collection, and cell isolation methodologies. They can maintain their phenotype longer in culture, a greater proliferative capacity, and a demonstrated therapeutic potential; in addition, they secrete a wide range of cytokines, GFs, macromolecules, and miRNAs directly into the surrounding micro-environment or through the microvesicles [168,174]; these bioactive factors exert various ‘trophic effects’ such as suppressing the local immune system, inhibiting apoptosis and scar formation (fibrosis), enhancing angiogenesis, stimulating mitosis, and inducing the differentiation of tissue-intrinsic reparative or stem cells [175]. ADSCs can be isolated more easily, using a much safer approach, in considerably larger amounts, are not only equally effective but may be much better suited than BM-MSCs in certain cases, in clinical applications [176].

The quality of ADSCs varies among donors based on their demographic profiles, such as age, gender, ethnicity, disease status, and body mass index [168]. The yield of ADSCs is 40 times higher than that of BMSCs, with a success rate of 100%, which may not decrease with age, making this type of tissue attractive for isolating MSCs and progenitor cells (PCs) [177]. However, the number, proliferative capacity, and ability of these cells to differentiate into multiple lineages reduced with age while cell senescence increased [178]. ADSCs functions are not limited to tissue-specific PCs but have multiple therapeutic effects mediated through paracrine and regulating angiogenesis signaling, inflammation, cell survival, cell homing, and other processes regarded as action mechanisms [179]. ADSCs have been used as a therapeutic agent in treating diabetes mellitus, corneal, articular cutaneous lesions, and liver disease, and repair of damaged cardiac tissues, and for developing novel therapeutic methods useful in reconstructive or tissue engineering [180,181].

ADSCs are a valuable therapeutic alternative for tissue rescue and repair due to their easy availability, immunomodulatory effects, and capacities for secretion of pro-angiogenesis and anti-apoptotic factors, differentiation into multiline age cells, and expansion [182]. The ADSCs secretomes modify tissue biology. Thus, exciting tissue-resident stem cells change immune cell activity and facilitate therapeutic outcomes [168]. They participate in modulating the changes caused by macrophages in the inflammatory phenotype, endorsing neo-angiogenesis mediated through ECs increased differentiation and migration and augmenting granulation tissue formation, ECM, and skin cells at proliferation and remodeling stages of wound healing that is imperative and relevant in designing innovative therapeutic strategies in regenerative medicine domain [183]. The regenerative tissue effects of ADSCs in vivo rely on an interaction between the soluble factors released by them and the recipient’s secretomes [184].

ADSCs may help repair tissue damage and help in neovascularization as a part of angiogenic therapy, as they can interact with and transform the wound-resident cells into matrix-building cells. This procedure is crucial for the dermal rebuilding course and epithelialization achieved through stimulating keratinocytes; a research study has elucidated ADSCs functioning as pericytes (in situ), facilitating vascular stability, and responding to environmental stimuli by communicating with ECs [107]. ADSCs may be vascular stem cells residing in a perivascular location and differentiate into smooth muscle and ECs used during angiogenesis and neo-vasculogenesis [185]. Coordination between ADSCs and ECs is required for network formation as ADSCs stabilize EC networks by enhancing pericyte-like characteristics. ADSCs induce vessel growth by secreting pro-angiogenic and regulatory proteins [186]. Induction in the expression of activin A is associated with new vessel formation. It directs the crosstalk between ADSCs and ECs, affecting these cell types of activity [187].

hADSCs produce exosomes that can induce cutaneous repair by regulating the remodeling of the ECM [188]. ADSCs-based cell therapies address wound healing of recalcitrant and chronic wounds by achieving full wound epithelialization rapidly and are safe without adverse effects for patients [107]. Various gene-modification approaches have been employed for manipulating genes and in vitro ADSCs preconditioning to increase the trophic factors production upon cell delivery in vivo [189]. In the past, ADSCs have been used to secrete VEGF in larger quantities to improve angiogenesis ability in therapeutic application in ischemic tissue [190].

## 7. Interactions between IL-2, IL-6, CD31, and CD34 with PRP and/or ADSCs in Use with Absorbable Meshes

Allogenic ADSCs and PRF combination can expedite full-thickness cartilage defect regeneration in the rabbit ear model devoid of any prominent immune trigger, as suggested by the lack of any prominent differences in the expression of the IL-2 gene in comparison with the control group [69]. IL-6 controls the healing process, especially in skin wounds, through migration, proliferation, and differentiation of stem cells, thus improving the healing process in skin wounds [191]. In response to inflammatory stimuli, ADSCs can produce IL-6 [192]. IL-6 released from ADSCs promoted the recovery of blood supply in the wounds [193]. The overexpression of S100A8, a calcium and zinc binding protein in ADSCs, significantly promoted their proliferation and differentiation, but the serum levels of IL-6 were significantly reduced, suggesting that S100A8 promoted the proliferation of ADSCs and inhibited inflammation to improve skin healing [194]. In in vivo studies, hematopoietic prostaglandin D synthase (HPGDS) was overexpressed to engineer hADSCs into hADSChpgds, and their effects on diabetic wound healing were evaluated using a full-thickness skin wound model in mice. The expression levels of IL-6 were prominently diminished. Still, the number of CD31+ ECs and scattered small blood vessels were significantly higher, indicating increased angiogenesis and vascularity in hADSChpgds group mice compared to the control [195]. In treating induced patellar tendon defects in rabbits with a PRP gel, the extent of neovascularization was significantly greater, as indicated by escalated expression of CD31 [196]. However, numerous studies have confirmed that ADSCs can express CD34 [197] but not CD31 [198]. Combined ADSCs and PRP therapy induced a strong angiogenic effect in diabetic albino rats as indicated by enhanced CD31 immuno-expression compared to control [199]. The in vitro co-culture of keloid tissue with ADSCs-CM brought a significant decrease in CD31+ and CD34+ vessels, thus exerting an anti-scarring effect [200]. Immuno-expression of CD31 in the endothelial cell lining of dermal blood vessels was enhanced in skin wounds in rats treated with ADSCs [201]. Wound healing studies in diabetic rats revealed that CD31+ cells were not only detected in the neo-capillaries, indicating spontaneous differentiation of engrafted ADSCs into vascular ECs, but also increased continuously and were detected in mature blood vessels, indicating a significant promotion of neovascularization of wounds [202]. In a study, topical application of ADSCs on excisional wounds on rabbit ears in full thickness resulted in enhanced expression of CD31 in granulation tissue CD31+ cells increased in the wound bed; however, CD31 expression in transplanted ADSCs was absently implicating a lack of ability to differentiate directly into ECs; similarly, CD34 expression was absent in ADSCs [203]. On the contrary, immuno-expression of CD31 in the endothelial cell lining of dermal blood vessels was enhanced in skin wounds in rats treated with ADSCs [188]. Wound healing studies in diabetic rats revealed that CD31+ cells were not only detected in the neo-capillaries, indicating spontaneous differentiation of engrafted ADSCs into vascular ECs, but also increased continuously and were detected in mature blood vessels, indicating a significant promotion of neovascularization of wounds [204]. In vitro co-culture of PRP and conditioned medium (CM) from ADSCs significantly stimulated the proliferation and migration of fibroblasts and keratinocytes, suggesting that PRP and ADSCs in combination may enhance healing and re-epithelialization of chronic wounds in vivo [205].

Alginate hydrogel containing EXOs derived from ADSCs productively increased wound closure, collagen synthesis, and vessel development, as demonstrated by the highest levels of CD31 expression compared to controls [206]. Velgraft^®^, a gelatin and chitosan biopolymer enhanced with ADSCs, improved wound healing by accelerating wound closure, rapid collagen synthesis, and deposition, thus leading to re-vascularization and re-epithelization. This was demonstrated through immunostaining, where CD31 positive expression was reported in ECs of neo-capillaries [207]. Three-dimensional scaffolds used in tissue engineering finely mimic the in vivo microenvironment and thus facilitate ADSCs’ localization, attachment, proliferation, and differentiation, suggesting that tissue-engineered ADSCs can substitute tissue and organ transplantation [208]. Three-dimensional cultivation using a collagen sponge scaffold promoted the differentiation of CD34-hADSCs into ECs, which may be applied as an artificial dermis to heal skin wounds [209]. ADSCs differentiate rapidly into ECs to form simple vessel-like structures in Matrigel^®^ substrates and thus may be crucial in regulating neo-vasculogenesis [210].

hADSCs stimulate wound healing in diabetic patients and function as a combined carrier scaffold for scar-less cutaneous repair [209]. For studying tissue repair in a murine skin injury model, two different sets of ECM scaffolds were used, namely-small intestinal submucosa (SIS) and acellular dermal matrix (ADM) and composite collagen–chondroitin sulfate–hyaluronic acid (Co–CS–HA) scaffold; the ADSCs-seeded scaffolds demonstrated enhanced wound healing capacity compared to non-seeded scaffolds; this suggested that ADSCs could be used as a source of cells to promote the vascularization capacities of scaffolds and that both ADSCs and the scaffolds exhibited synergistic effects in promoting angiogenesis; moreover, some ADSCs demonstrated GFP co-localization with CD31 implicating spontaneous differentiation into a vascular endothelial phenotype; ADSCs were negative for CD34 [210]. A bioactive PRP scaffold capable of releasing endogenous GFs, BMSCs, and ADSCs to differentiate into chondrocytes may be suitable for cell-based cartilage repair [211]. In a skin graft study in rats, the number of subcutaneous, neovascular CD31+ cells in the ADSCs embedded in the PRP gel scaffold-treated group was significantly greater than the control group, indicating that ADSCs+ PRP significantly induced early stage neovascularization after skin graft transplantation [212].

PRP stimulates the proliferation and differentiation, maintains the multipotency of MSCs, promotes their migration into the wound area, and enhances the wound-healing pathway; PRP or MSCs, and MSCs + PRP improved and accelerated wound healing represent a potential therapeutic approach [211]. PRP-gels can function as carriers for delivering both human MSCs and GFs in tissue engineering; the platelet concentration of PRP is crucial in providing the most favorable microenvironment for MSCs concerning the clinical application of PRP-gels [211].

## 8. Recent Advances and Challenges

Wound healing involving absorbable meshes constitutes a central area of focus in both surgical and medical research. This domain centers its endeavors on the manipulation of mesh materials to enhance and streamline tissue healing across a spectrum of wound types. These absorbable meshes are intentionally engineered to offer mechanical reinforcement during the initial phases of wound healing. As time progresses, they gradually degrade in synchronization with the tissue’s recovery process. This property eliminates the necessity for subsequent removal procedures. In the course of research, significant attention has been directed toward understanding the intricate interaction between absorbable meshes and the body’s intrinsic healing mechanisms. This interaction exerts a noticeable influence on the complex process of wound repair. Notable progress has been achieved in the design of absorbable mesh materials to ensure biocompatibility [213]. This ensures the seamless integration of these materials within the body, thereby preventing disruptive inflammatory reactions. Additionally, absorbable meshes that can integrate with neighboring tissues have provided insights into facilitating the formation of new blood vessels and promoting the migration of cells to the injury site. This integration is vital for enduring wound healing and the mitigation of potential complications. Furthermore, researchers have devoted their efforts to finely-tuning the controlled degradation of these meshes [214]. This controlled degradation allows for mechanical strength during the crucial early stages of wound healing. 

However, these significant advancements have encountered challenges and variables that require thorough consideration. A prominent concern is the susceptibility to infections. Absorbable meshes, being foreign elements, could exacerbate the risk of infections if not managed judiciously [215]. Striking a delicate balance between the mechanical properties of the mesh and its rate of degradation represents another challenge. If the degradation process occurs too rapidly, it could compromise essential support during critical phases of healing. Conversely, prolonged degradation might lead to complications or necessitate intervention. Furthermore, the variability in individual responses to absorbable meshes introduces complexity in predicting outcomes. This variability arises from distinct immune reactions, overall health statuses, and genetic compositions. Ensuring the sustained effectiveness of absorbable meshes in wound repair necessitates prolonged observation of patient outcomes. This extended observation is crucial to elucidate their effects on wound healing, scarring, and recurrence rates.

## 9. Conclusions

Wound healing is a multifaceted process involving intricate interactions among various elements. Cellular responses, molecular factors, and the introduction of biomaterials come together in a delicate interplay to orchestrate the repair of damaged tissue. The process of wound healing can be broken down into distinct phases: inflammation, proliferation, and remodeling. In the inflammatory phase, immune cells are recruited to the site of injury to manage any potential infections and begin the clean-up process. During proliferation, growth factors stimulate the production of new cells and blood vessels, aiding in the reconstruction of the damaged tissue. Remodeling involves refining the tissue’s structure for improved strength and function.
Immune cells, growth factors, and extracellular matrix components are central to the wound-healing process. Immune cells help clear debris and pathogens, while growth factors stimulate cell division and tissue regrowth. Extracellular matrix components provide the structural framework for new tissue formation.The involvement of cytoskeletal elements and proteins like the ‘Formin’ family shed light on the intricate cellular mechanisms driving wound healing. These mechanisms contribute to cell migration, proliferation, and tissue reorganization during the healing process.Incorporating absorbable meshes into wound repair strategies presents a promising avenue for enhancing healing outcomes. These meshes act as scaffolds that support the regeneration of tissue. They facilitate interactions with growth factors, cytokines, and various cellular components, such as interleukins, CD31, CD34, platelet-rich plasma (PRP), and adipose-derived stem cells (ADSCs). These interactions have the potential to expedite and optimize the healing process.

Particularly for individuals with diabetes, who often face compromised wound healing, absorbable meshes offer a renewed sense of optimism. These meshes can be engineered to release healing factors in a controlled manner, addressing the challenges that diabetic patients often encounter in the wound healing process.

Despite significant progress in wound healing research, several challenges persist. Ensuring the effective integration of meshes, maintaining biocompatibility, preventing infections, and managing postoperative complications are areas that continue to demand focused research and innovative solutions. Advances in mesh design, placement techniques, and personalized approaches are pivotal to shaping the field of wound healing interventions. Taking into account patient-specific factors ensures tailored solutions that can optimize outcomes and minimize risks. The ever-evolving nature of wound healing research necessitates collaboration among researchers, clinicians, and biomaterial scientists. By working together, they can unravel the complexities of wound healing processes and develop strategies that cater to diverse patient populations.

## Data Availability

Not applicable.

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
