# Peer review of "The Effects of Tissue Healing Factors in Wound Repair Involving Absorbable Meshes: A Narrative Review"

_jcm, 2023, doi:10.3390/jcm12175683_

Round 1
Reviewer 1 Report
The written work presents a detailed overview of tissue healing factors as well as meshes. Relevant and recent literature has been extensively cited and included.
The abstract represents one of the weaknesses of this work. It is unnecessarily short (130 words out of 200 allowed). For example, the teased three-step process of wound healing can be mentioned in keywords. Furthermore, the abstract gives little specific information (see lines 15-17: "Various types of meshes, such as non-absorbable and absorbable synthetic and absorbable biological, are available for facilitating wound repair"). This is not very informative and common knowledge. Perhaps a more reflective approach could be taken instead.
The chapter on meshes in the introduction provides an overview of the existing meshes, their types and applications.
The large number of references has not been thoroughly incorporated into the text at certain points. For example, the following passage in lines 62/63: "Three categories of meshes are currently available for wound repair: non-absorbable synthetic meshes, absorbable synthetic meshes, and absorbable biological meshes [15]. In reference 15, entitled "3D TECA hydrogel reduces cellular senescence and enhances fibroblasts migration in wound healing", the effect of an inflammatory mark on periodontal ligament fibroblasts (PDLFs) was investigated in vitro. Biological meshes are not mentioned in this publication.
Further, the authors have a surgical background, so a more differentiated view of the use of these meshes to illuminate the practical approach would be useful. In particular, postoperative complications (including minor complications such as infections or seromas) should be addressed.
Overall, the review depicts the function of tissue healing factors in great detail.
However, the interaction with meshes is presented weakly. Point 2 is called "Role of various tissue healing factors in association with absorbable meshes". Point a deals with IL-2, but the association with absorbable meshes is not adequately elaborated. The same is true for IL-6.
Side note: Point 2 could be named more specifically: it is called role of various tissue healing factors and then describes two tissue healing factors IL-2 and IL-6.
Further the bridge between tissue healing factors and meshes should find more resonance in the conclusion section. More precisely, the hypothesis “However, studies regarding the applicability of these six factors in combination with each other when used with absorbable meshes for tissue repair are exceedingly rare. Hence, this review of the literature was undertaken to address this lacuna.” was not revisited in the conclusions, which is great limitation of this work. Meshes are not even mentioned in the conclusions, which means that the claim "to fill the supposed lacuna in the literature" was not made.
Minor issues:
When specific medical products are mentioned, the manufacturer's name should be indicated with City and State. (e.g. Line 78)
The large number of typos and formatting errors is extremely noticeable. Examples: the Introduction was marked with "1. .Introduction". From point 3 on, the bold font was omitted. The Conclusions section has not been numbered at all. All points between Intruction and Conclusion end with "... in/for wound healing" perhaps this ending could be omitted at all points to create a cleaner look. Something that can be addressed later, but which clearly disturbs the flow of reading, is the extensive omission of blank lines after parentheses, commas and periods.
Author Response
The authors are immensely grateful to the reviewers for their comments which certainly enhances the manuscript. The following are responses to each comments posed by the reviewers.
Reviewer 1:
The written work presents a detailed overview of tissue healing factors as well as meshes. Relevant and recent literature has been extensively cited and included.
- The abstract represents one of the weaknesses of this work. It is unnecessarily short (130 words out of 200 allowed). For example, the teased three-step process of wound healing can be mentioned in keywords. Furthermore, the abstract gives little specific information (see lines 15- 17: "Various types of meshes, such as non-absorbable and absorbable synthetic and absorbable biological, are available for facilitating wound repair"). This is not very informative and common knowledge. Perhaps a more reflective approach could be taken instead.
Response: The abstract has been rewritten.
- The chapter on meshes in the introduction provides an overview of the existing meshes, their types and applications. The large number of references has not been thoroughly incorporated into the text at certain points. For example, the following passage in lines 62/63: "Three categories of meshes are currently available for wound repair: non-absorbable synthetic meshes, absorbable synthetic meshes, and absorbable biological meshes [15]. In reference 15, entitled "3D TECA hydrogel reduces cellular senescence and enhances fibroblasts migration in wound healing", the effect of an inflammatory mark on periodontal ligament fibroblasts (PDLFs) was investigated in vitro. Biological meshes are not mentioned in this publication.
Response: The citation has been replaced with a relevant one. All references in this section has been has been rechecked for relevance.
- Further, the authors have a surgical background, so a more differentiated view of the use of these meshes to illuminate the practical approach would be useful. In particular, postoperative complications (including minor complications such as infections or seromas) should be addressed.
Response: The following text is added in the introduction section:
The selection of an appropriate mesh type is guided by patient-specific factors, tissue characteristics, and the nature of the surgery. Optimal mesh integration with surrounding tissues is ensured through precise placement techniques, meticulous fixation, and proper sizing.
Despite the advantages, attention needs to be given to postoperative complications of mesh integration such as infections and seromas. Infections can manifest around the mesh, giving rise to symptoms such as redness, swelling, pain, and fever. Addressing these infections might necessitate antibiotic treatment and, in severe instances, even mesh removal (Kao AM, Arnold MR, Augenstein VA, Heniford BT. Prevention and treatment strategies for mesh infection in abdominal wall reconstruction. Plastic and reconstructive surgery. 2018 Sep 1;142(3S):149S-55S.). Additionally, the formation of seromas can impede wound healing and heighten infection susceptibility (Mayagoitia JC, Almaraz A, Diaz C. Two cases of cystic seroma following mesh incisional hernia repair. Hernia. 2006 Mar;10(1):83-6.). Chronic pain at the mesh site, attributed to nerve irritation or entrapment, is observed in certain patients, thereby highlighting the importance of prudent mesh selection, precise placement, and surgical methodology to mitigate this concern (Bendavid R, Lou W, Grischkan D, Koch A, Petersen K, Morrison J, Iakovlev V. A mechanism of mesh-related post-herniorrhaphy neuralgia. Hernia. 2016 Jun;20:357-65.). Despite mesh implementation, the occurrence of wound dehiscence remains possible. However, meticulous tension management, adherence to postoperative directives, and comprehensive patient education regarding activity limitations stand as crucial measures in averting this complication (Lima HV, Rasslan R, Novo FC, Lima TM, Damous SH, Bernini CO, Montero EF, Utiyama EM. Prevention of fascial dehiscence with onlay prophylactic mesh in emergency laparotomy: a randomized clinical trial. Journal of the American College of Surgeons. 2020 Jan 1;230(1):76-87.).
- Overall, the review depicts the function of tissue healing factors in great detail. However, the interaction with meshes is presented weakly. Point 2 is called "Role of various tissue healing factors in association with absorbable meshes". Point a deals with IL-2, but the association with absorbable meshes is not adequately elaborated. The same is true for IL-6. Side note: Point 2 could be named more specifically: it is called role of various tissue healing factors and then describes two tissue healing factors IL-2 and IL-6.
Response: The title of section 2 has been changed to “Role of Interleukines in wound healing’ to better reflect the content of the section.
- Further the bridge between tissue healing factors and meshes should find more resonance in the conclusion section. More precisely, the hypothesis "However, studies regarding the applicability of these six factors in combination with each other when used with absorbable meshes for tissue repair are exceedingly rare. Hence, this review of the literature was undertaken to address this lacuna." was not revisited in the conclusions, which is great limitation of this work. Meshes are not even mentioned in the conclusions, which means that the claim "to fill the supposed lacuna in the literature" was not made.
Response: The following paragraph has been added in the beginning of section 2 to address the above gap and discuss the interaction of interleukins and meshes. Likewise the conclusion section has been modified to better summarize the article.
Interleukins (ILs) are a group of signaling molecules that play a crucial role in the immune response, inflammation, and tissue repair processes, including wound healing. The wound healing process begins with inflammation, during which immune cells are recruited to the wound site. Interleukins, such as IL-1, IL-6, and IL-8, play a significant role in initiating and regulating the inflammatory response. These cytokines promote the migration of immune cells to the wound site and help in removing debris and pathogens. These interleukins might influence the recruitment of immune cells to the mesh site and contribute to the breakdown of the mesh material. The mesh material itself can trigger an immune response, leading to the secretion of various interleukins and other cytokines. The presence of these interleukins can influence the recruitment of immune cells, the proliferation of fibroblasts, and the remodeling of tissue around the mesh. Proper modulation of interleukin activity is essential to ensure that the healing process occurs without excessive inflammation or fibrosis.
- Minor issues: When specific medical products are mentioned, the manufacturer's name should be indicated with City and State. (e.g. Line 78) The large number of typos and formatting errors is extremely noticeable. Examples: the Introduction was marked with "1. .Introduction". From point 3 on, the bold font was omitted.
- The Conclusions section has not been numbered at all. All points between Intruction and Conclusion end with "... in/for wound healing" perhaps this ending could be omitted at all points to create a cleaner look. Something that can be addressed later, but which clearly disturbs the flow of reading, is the extensive omission of blank lines after parentheses, commas and periods.
Response: Wherever applicable, the phrase "... in/for wound healing" has been omitted as suggested.
Reviewer 2 Report
Dear Author
Article entitled The effects of tissue healing factors in wound repair involving
absorbable meshes: a narrative review
is of interest in scienfic community.
However below mentioned points need to be consider for possible considertaion for publication
1. Abstract need to rewrite (end part) with respect to conisideration of overall effects/impact and what precisely this revoew article focus on.
2. Diagramatic representation/llustration is highly recommended in introduction
3. Mechanistic/ Mechanism based illustration is highly recommended for each of 6 factors
4. Other factors like psychological, diabetic etc may be explore
5. Challanges and approaches/recent advancements to overcome may be explore
Minor editing of English language required
Author Response
The authors are immensely grateful to the reviewers for their comments which certainly enhances the manuscript. The following are responses to each comments posed by the reviewers.
Reviewer 2:
Article entitled The effects of tissue healing factors in wound repair involving absorbable meshes: a narrative review is of interest in scienfic community. However below mentioned points need to be consider for possible considertaion for publication
- Abstract need to rewrite (end part) with respect to conisideration of overall effects/impact and what precisely this revoew article focus on.
Response: The abstract has been rewritten.
- Diagramatic representation/llustration is highly recommended in introduction
- Mechanistic/ Mechanism based illustration is highly recommended for each of 6 factors
- Other factors like psychological, diabetic etc may be explore
Response: The following information is added in the introduction section.
Furthermore, diabetic individuals often face challenges in wound healing due to compromised blood circulation and reduced immune response. Absorbable meshes, serving as scaffolds, introduce a novel dimension to the healing process (Brem H, Tomic-Canic M. Cellular and molecular basis of wound healing in diabetes. The Journal of clinical investigation. 2007 May 1;117(5):1219-22.). These meshes not only provide mechanical support to the wound site but also act as vehicles for controlled release of growth factors and cytokines, thereby fostering a conducive environment for tissue regeneration. By influencing aspects like angiogenesis, collagen synthesis, and cell migration, these healing factors within the context of absorbable meshes can significantly expedite wound closure and minimize the risk of infection, offering renewed hope for enhanced recovery in diabetic patients grappling with chronic wounds.
- Challanges and approaches/recent advancements to overcome may be explore
Response: The following information has been added as a new section in the article.
8. Recent advances and challenges
Wound healing involving absorbable meshes constitutes a central area of focus in both surgical and medical research. This domain centers its endeavors on the manipulation of mesh materials to enhance and streamline tissue healing across a spectrum of wound types. These absorbable meshes are intentionally engineered to offer mechanical reinforcement during the initial phases of wound healing. As time progresses, they gradually degrade in synchronization with the tissue's recovery process. This property eliminates the necessity for subsequent removal procedures. In the course of research, significant attention has been directed toward understanding the intricate interaction between absorbable meshes and the body's intrinsic healing mechanisms. This interaction exerts a noticeable influence on the complex process of wound repair. Notable progress has been achieved in the design of absorbable mesh materials to ensure biocompatibility (Lucchina AG, Radica MK, Costa AL, Mortellaro C, Soliani G, Zavan B. Mesh-tissue integration of synthetic and biologic meshes in wall surgery: brief state of art. European Review for Medical & Pharmacological Sciences. 2022 Feb 2;26.). This ensures the seamless integration of these materials within the body, thereby preventing disruptive inflammatory reactions. Additionally, absorbable meshes can integrate with neighboring tissues has provided insights into facilitating the formation of new blood vessels and promoting the migration of cells to the injury site. This integration is vital for enduring wound healing and the mitigation of potential complications. Furthermore, researchers have devoted their efforts to finely-tuning the controlled degradation of these meshes (Khandaker M, Alkadhem N, Progri H, Nikfarjam S, Jeon J, Kotturi H, Vaughan MB. Glutathione immobilized polycaprolactone nanofiber mesh as a dermal drug delivery mechanism for wound healing in a diabetic patient. Processes. 2022 Mar 4;10(3):512.). This controlled degradation allows for mechanical strength during the crucial early stages of wound healing.
However, these significant advancements have encountered challenges and variables that require thorough consideration. A prominent concern is the susceptibility to infections. Absorbable meshes, being foreign elements, could exacerbate the risk of infections if not managed judiciously (Xu D, Fang M, Wang Q, Qiao Y, Li Y, Wang L. Latest trends on the attenuation of systemic foreign body response and infectious complications of synthetic hernia meshes. ACS Applied Bio Materials. 2021 Dec 14;5(1):1-9.). Striking a delicate balance between the mechanical properties of the mesh and its rate of degradation represents another challenge. If the degradation process occurs too rapidly, it could compromise essential support during critical phases of healing. Conversely, a prolonged degradation might lead to complications or necessitate intervention. Furthermore, the variability in individual responses to absorbable meshes introduces complexity in predicting outcomes. This variability arises from distinct immune reactions, overall health statuses, and genetic compositions. Ensuring the sustained effectiveness of absorbable meshes in wound repair necessitates prolonged observation of patient outcomes. This extended observation is crucial to elucidate their effects on wound healing, scarring, and recurrence rates.
- Comments on the Quality of English Language: Minor editing of English language required
Response: The manuscript has been thoroughly checked for English grammar and syntax.